# GRAPH INFERENCE LEARNING FOR SEMI-SUPERVISED CLASSIFICATION

**Chunyan Xu, Zhen Cui**[*]**, Xiaobin Hong, Tong Zhang, and Jian Yang**
School of Computer Science and Engineering, Nanjing University of Science and Technology, Nanjing, China
`{cyx,zhen.cui,xbhong,tong.zhang,csjyang}@njust.edu.cn`

**Wei Liu**
Tencent AI Lab, China
`wl2223@columbia.edu`

## ABSTRACT

In this work, we address semi-supervised classification of graph data, where the categories of those unlabeled nodes are inferred from labeled nodes as well as graph structures. Recent works often solve this problem via advanced graph convolution in a conventionally supervised manner, but the performance could degrade significantly when labeled data is scarce. To this end, we propose a Graph Inference Learning (GIL) framework to boost the performance of semi-supervised node classification by learning the inference of node labels on graph topology. To bridge the connection between two nodes, we formally define a structure relation by encapsulating node attributes, between-node paths, and local topological structures together, which can make the inference conveniently deduced from one node to another node. For learning the inference process, we further introduce meta-optimization on structure relations from training nodes to validation nodes, such that the learnt graph inference capability can be better self-adapted to testing nodes. Comprehensive evaluations on four benchmark datasets (including Cora, Citeseer, Pubmed, and NELL) demonstrate the superiority of our proposed GIL when compared against state-of-the-art methods on the semi-supervised node classification task.

## 1 INTRODUCTION

Graph, which comprises a set of vertices/nodes together with connected edges, is a formal structural representation of non-regular data. Due to the strong representation ability, it accommodates many potential applications, e.g., social network (Orsini et al., 2017), world wide data (Page et al., 1999), knowledge graph (Xu et al., 2017), and protein-interaction network (Borgwardt et al., 2007). Among these, semi-supervised node classification on graphs is one of the most interesting also popular topics. Given a graph in which some nodes are labeled, the aim of semi-supervised classification is to infer the categories of those remaining unlabeled nodes by using various priors of the graph.

While there have been numerous previous works (Brandes et al., 2008; Zhou et al., 2004; Zhu et al., 2003; Yang et al., 2016; Zhao et al., 2019) devoted to semi-supervised node classification based on explicit graph Laplacian regularizations, it is hard to efficiently boost the performance of label prediction due to the strict assumption that connected nodes are likely to share the same label information. With the progress of deep learning on grid-shaped images/videos (He et al., 2016), a few of graph convolutional neural networks (CNN) based methods, including spectral (Kipf & Welling, 2017) and spatial methods (Niepert et al., 2016; Pan et al., 2018; Yu et al., 2018), have been proposed to learn local convolution filters on graphs in order to extract more discriminative node representations. Although graph CNN based methods have achieved considerable capabilities of graph embedding by optimizing filters, they are limited into a conventionally semi-supervised framework and lack of an efficient inference mechanism on graphs. Especially, in the case of few-shot learning, where a small number of training nodes are labeled, this kind of methods would drastically compromise the performance. For example, the Pubmed graph dataset (Sen et al., 2008) consists

---

[*]Corresponding author: Zhen Cui.

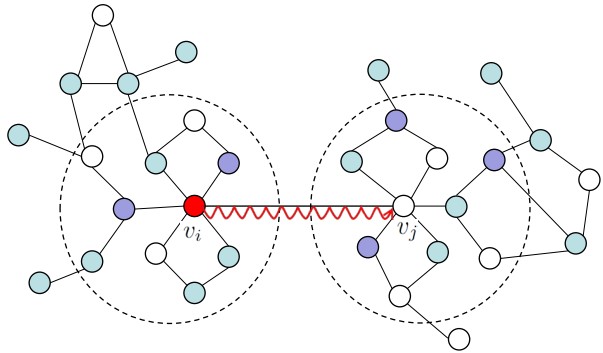

Figure 1: The illustration of our proposed GIL framework. For the problem of graph node labeling, the category information of these unlabeled nodes depends on the similarity computation between a query node (e.g., $v_j$) and these labeled reference nodes (e.g., $v_i$). We consider the similarity from three points: node attributes, the consistency of local topological structures (i.e., the circle with dashed line), and the between-node path reachability (i.e., the red wave line from $v_i$ to $v_j$). Specifically, the local structures as well as node attributes are encoded as high-level features with graph convolution, while the between-node path reachability is abstracted as reachable probabilities of random walks. To better make the inference generalize to test nodes, we introduce a meta-learning strategy to optimize the structure relations learning from training nodes to validation nodes.

of 19,717 nodes and 44,338 edges, but only 0.3% nodes are labeled for the semi-supervised node classification task. These aforementioned works usually boil down to a general classification task, where the model is learnt on a training set and selected by checking a validation set. However, they do not put great efforts on how to learn to infer from one node to another node on a topological graph, especially in the few-shot regime.

In this paper, we propose a graph inference learning (GIL) framework to teach the model itself to adaptively infer from reference labeled nodes to those query unlabeled nodes, and finally boost the performance of semi-supervised node classification in the case of a few number of labeled samples. Given an input graph, GIL attempts to infer the unlabeled nodes from those observed nodes by building between-node relations. The between-node relations are structured as the integration of node attributes, connection paths, and graph topological structures. It means that the similarity between two nodes is decided from three aspects: the consistency of node attributes, the consistency of local topological structures, and the between-node path reachability, as shown in Fig. 1. The local structures anchored around each node as well as the attributes of nodes therein are jointly encoded with graph convolution (Defferrard et al., 2016) for the sake of high-level feature extraction. For the between-node path reachability, we adopt the random walk algorithm to obtain the characteristics from a labeled reference node $v_i$ to a query unlabeled node $v_j$ in a given graph. Based on the computed node representation and between-node reachability, the structure relations can be obtained by computing the similar scores/relationships from reference nodes to unlabeled nodes in a graph. Inspired by the recent meta-learning strategy (Finn et al., 2017), we learn to infer the structure relations from a training set to a validation set, which can benefit the generalization capability of the learned model. In other words, our proposed GIL attempts to learn some transferable knowledge underlying in the structure relations from training samples to validation samples, such that the learned structure relations can be better self-adapted to the new testing stage.

We summarize the main contributions of this work as three folds:

- We propose a novel graph inference learning framework by building structure relations to infer unknown node labels from those labeled nodes in an end-to-end way. The structure relations are well defined by jointly considering node attributes, between-node paths, and graph topological structures.

- To make the inference model better generalize to test nodes, we introduce a meta-learning procedure to optimize structure relations, which could be the first time for graph node classification to the best of our knowledge.

- Comprehensive evaluations on three citation network datasets (including Cora, Citeseer, and Pubmed) and one knowledge graph data (i.e., NELL) demonstrate the superiority of our proposed GIL in contrast with other state-of-the-art methods on the semi-supervised classification task.

## 2 RELATED WORK

**Graph CNNs:** With the rapid development of deep learning methods, various graph convolution neural networks (Kashima et al., 2003; Morris et al., 2017; Shervashidze et al., 2009; Yanardag & Vishwanathan, 2015; Jiang et al., 2019; Zhang et al., 2020) have been exploited to analyze the irregular graph-structured data. For better extending general convolutional neural networks to graph domains, two broad strategies have been proposed, including spectral and spatial convolution methods. Specifically, spectral filtering methods (Henaff et al., 2015; Kipf & Welling, 2017) develop convolution-like operators in the spectral domain, and then perform a series of spectral filters by decomposing the graph Laplacian. Unfortunately, the spectral-based approaches often lead to a high computational complex due to the operation of eigenvalue decomposition, especially for a large number of graph nodes. To alleviate this computation burden, local spectral filtering methods (Defferrard et al., 2016) are then proposed by parameterizing the frequency responses as a Chebyshev polynomial approximation. Another type of graph CNNs, namely spatial methods (Li et al., 2016; Niepert et al., 2016), can perform the filtering operation by defining the spatial structures of adjacent vertices. Various approaches can be employed to aggregate or sort neighboring vertices, such as diffusion CNNs (Atwood & Towsley, 2016), GraphSAGE (Hamilton et al., 2017), PSCN (Niepert et al., 2016), and NgramCNN (Luo et al., 2017). From the perspective of data distribution, recently, the Gaussian induced convolution model (Jiang et al., 2019) is proposed to disentangle the aggregation process through encoding adjacent regions with Gaussian mixture model.

**Semi-supervised node classification:** Among various graph-related applications, semi-supervised node classification has gained increasing attention recently, and various approaches have been proposed to deal with this problem, including explicit graph Laplacian regularization and graph-embedding approaches. Several classic algorithms with graph Laplacian regularization contain the label propagation method using Gaussian random fields (Zhu et al., 2003), the regularization framework by relying on the local/global consistency (Zhou et al., 2004), and the random walk-based sampling algorithm for acquiring the context information (Yang et al., 2016). To further address scalable semi-supervised learning issues (Liu et al., 2012), the Anchor Graph regularization approach (Liu et al., 2010) is proposed to scale linearly with the number of graph nodes and then applied to massive-scale graph datasets. Several graph convolution network methods (Abu-El-Haija et al., 2018; Du et al., 2017; Thekumparampil et al., 2018; Velickovic et al., 2018; Zhuang & Ma, 2018) are then developed to obtain discriminative representations of input graphs. For example, Kipf et al. (Kipf & Welling, 2017) proposed a scalable graph CNN model, which can scale linearly in the number of graph edges and learn graph representations by encoding both local graph structures and node attributes. Graph attention networks (GAT) (Velickovic et al., 2018) are proposed to compute hidden representations of each node for attending to its neighbors with a self-attention strategy. By jointly considering the local- and global-consistency information, dual graph convolutional networks (Zhuang & Ma, 2018) are presented to deal with semi-supervised node classification. The critical difference between our proposed GIL and those previous semi-supervised node classification methods is to adopt a graph inference strategy by defining structure relations on graphs and then leverage a meta optimization mechanism to learn an inference model, which could be the first time to the best of our knowledge, while the existing graph CNNs take semi-supervised node classification as a general classification task.

## 3 THE PROPOSED MODEL

### 3.1 PROBLEM DEFINITION

Formally, we denote an undirected/directed graph as $\mathcal{G} = \{\mathcal{V}, \mathcal{E}, \mathcal{X}, \mathcal{Y}\}$, where $\mathcal{V} = \{v_i\}_{i=1}^n$ is the finite set of $n$ (or $|\mathcal{V}|$) vertices, $\mathcal{E} \in \mathbb{R}^{n \times n}$ defines the adjacency relationships (i.e., edges) between vertices representing the topology of $\mathcal{G}$, $\mathcal{X} \in \mathbb{R}^{n \times d}$ records the explicit/implicit attributes/signals of vertices, and $\mathcal{Y} \in \mathbb{R}^n$ is the vertex labels of $C$ classes. The edge $\mathcal{E}_{ij} = \mathcal{E}(v_i, v_j) = 0$ if and only if vertices $v_i, v_j$ are not connected, otherwise $\mathcal{E}_{ij} \neq 0$. The attribute matrix $\mathcal{X}$ is attached to the vertex set $\mathcal{V}$, whose $i$-th row $\mathcal{X}_{v_i}$ (or $\mathcal{X}_{i\cdot}$) represents the attribute of the $i$-th vertex $v_i$. It means that $v_i \in \mathcal{V}$ carries a vector of $d$-dimensional signals. Associated with each node $v_i \in \mathcal{V}$, there is a discrete label $y_i \in \{1, 2, \cdots, C\}$.

We consider the task of semi-supervised node classification over graph data, where only a small number of vertices are labeled for the model learning, i.e., $|\mathcal{V}_{Label}| \ll |\mathcal{V}|$. Generally, we have three node sets: a training set $\mathcal{V}_{tr}$, a validation set $\mathcal{V}_{val}$, and a testing set $\mathcal{V}_{te}$. In the standard protocol

of prior literatures (Yang et al., 2016), the three node sets share the same label space. We follow but do not restrict this protocol for our proposed method. Given the training and validation node sets, the aim is to predict the node labels of testing nodes by using node attributes as well as edge connections. A sophisticated machine learning technique used in most existing methods (Kipf & Welling, 2017; Zhou et al., 2004) is to choose the optimal classifier (trained on a training set) after checking the performance on the validation set. However, these methods essentially ignore how to extract transferable knowledge from these known labeled nodes to unlabeled nodes, as the graph structure itself implies node connectivity/reachability. Moreover, due to the scarcity of labeled samples, the performance of such a classifier is usually not satisfying. To address these issues, we introduce a meta-learning mechanism (Finn et al., 2017; Ravi & Larochelle, 2017; Sung et al., 2017) to learn to infer node labels on graphs. Specifically, the graph structure, between-node path reachability, and node attributes are jointly modeled into the learning process. Our aim is to learn to infer from labeled nodes to unlabeled nodes, so that the learner can perform better on a validation set and thus classify a testing set more accurately.

## 3.2 STRUCTURE RELATION

For convenient inference, we specifically build a structure relation between two nodes on the topology graph. The labeled vertices (in a training set) are viewed as the reference nodes, and their information can be propagated into those unlabeled vertices for improving the label prediction accuracy. Formally, given a reference node $v_i \in \mathcal{V}_{Label}$, we define the score of a query node $v_j$ similar to $v_i$ as

$$s_{i \to j} = f_r(f_e(\mathcal{G}_{v_i}), f_e(\mathcal{G}_{v_j}), f_{\mathcal{P}}(v_i, v_j, \mathcal{E})), \tag{1}$$

where $\mathcal{G}_{v_i}$ and $\mathcal{G}_{v_j}$ may be understood as the centralized subgraphs around $v_i$ and $v_j$, respectively. $f_e, f_r, f_{\mathcal{P}}$ are three abstract functions that we explain as follows:

- Node representation $f_e(\mathcal{G}_{v_i}) \longrightarrow \mathbb{R}^{d_v}$, encodes the local representation of the centralized subgraph $\mathcal{G}_{v_i}$ around node $v_i$, and may thus be understood as a local filter function on graphs. This function should not only take the signals of nodes therein as input, but also consider the local topological structure of the subgraph for more accurate similarity computation. To this end, we perform the spectral graph convolution on subgraphs to learn discriminative node features, analogous to the pixel-level feature extraction from convolution maps of gridded images. The details of feature extraction $f_e$ are described in Section 4.

- Path reachability $f_{\mathcal{P}}(v_i, v_j, \mathcal{E}) \longrightarrow \mathbb{R}^{d_p}$, represents the characteristics of path reachability from $v_i$ to $v_j$. As there usually exist multiple traversal paths between two nodes, we choose the function as reachable probabilities of different lengths of walks from $v_i$ to $v_j$. More details will be introduced in Section 4.

- Structure relation $f_r(\mathbb{R}^{d_v}, \mathbb{R}^{d_v}, \mathbb{R}^{d_p}) \longrightarrow \mathbb{R}$, is a relational function computing the score of $v_j$ similar to $v_i$. This function is not exchangeable for different orders of two nodes, due to the asymmetric reachable relationship $f_{\mathcal{P}}$. If necessary, we may easily revise it as a symmetry function, e.g., summarizing two traversal directions. The score function depends on triple inputs: the local representations extracted from the subgraphs w.r.t. $f_e(\mathcal{G}_{v_i})$ and $f_e(\mathcal{G}_{v_j})$, respectively, and the path reachability from $v_i$ to $v_j$.

In semi-supervised node classification, we take the training node set $\mathcal{V}_{tr}$ as the reference samples, and the validation set $\mathcal{V}_{val}$ as the query samples during the training stage. Given a query node $v_j \in \mathcal{V}_{val}$, we can derive the class similarity score of $v_j$ w.r.t. the $c$-th ($c = 1, \cdots, C$) category by weighting the reference samples $\mathcal{C}_c = \{v_k | y_{v_k} = c\}$. Formally, we can further revise Eqn. (1) and define the class-to-node relationship function as

$$s_{\mathcal{C}_c \to j} = \phi_r(F_{\mathcal{C}_c \to v_j} \sum_{v_i \in \mathcal{C}_c} w_{i \to j} \cdot f_e(\mathcal{G}_{v_i}), f_e(\mathcal{G}_{v_j})), \tag{2}$$

$$\text{s.t.} \quad w_{i \to j} = \phi_w(f_{\mathcal{P}}(v_i, v_j, \mathcal{E})), \tag{3}$$

where the function $\phi_w$ maps a reachable vector $f_{\mathcal{P}}(v_i, v_j, \mathcal{E})$ into a weight value, and the function $\phi_r$ computes the similar score between $v_j$ and the $c$-th class nodes. The normalization factor $F_{\mathcal{C}_c \to v_j}$ of the $c$-th category w.r.t. $v_j$ is defined as

$$F_{\mathcal{C}_c \to v_j} = \frac{1}{\sum_{v_i \in \mathcal{C}_c} w_{i \to j}}. \tag{4}$$

For the relation function $\phi_r$ and the weight function $\phi_w$, we may choose some subnetworks to instantiate them in practice. The detailed implementation of our model can be found in Section 4.

### 3.3 INFERENCE LEARNING

According to the class-to-node relationship function in Eqn. (2), given a query node $v_j$, we can obtain a score vector $\mathbf{s}_{\mathcal{C} \to j} = [s_{\mathcal{C}_1 \to j}, \cdots, s_{\mathcal{C}_C \to j}]^\intercal \in \mathbb{R}^C$ after computing the relations to all classes . The indexed category with the maximum score is assumed to be the estimated label. Thus, we can define the loss function based on cross entropy as follows:

$$\mathcal{L} = -\sum_{v_j} \sum_{c=1}^{C} y_{j,c} \log \hat{y}_{\mathcal{C}_c \to j}, \tag{5}$$

where $y_{j,c}$ is a binary indicator (i.e., 0 or 1) of class label $c$ for node $v_j$, and the softmax operation is imposed on $s_{\mathcal{C}_c \to j}$, i.e., $\hat{y}_{\mathcal{C}_c \to j} = \exp(s_{\mathcal{C}_c \to j}) / \sum_{k=1}^{C} \exp(s_{\mathcal{C}_k \to j})$. Other error functions may be chosen as the loss function, e.g., mean square error. In the regime of general classification, the cross entropy loss is a standard one that performs well.

Given a training set $\mathcal{V}_{tr}$, we expect that the best performance can be obtained on the validation set $\mathcal{V}_{val}$ after optimizing the model on $\mathcal{V}_{tr}$. Given a trained/pretrained model $\Theta = \{f_e, \phi_w, \phi_r\}$, we perform iteratively gradient updates on the training set $\mathcal{V}_{tr}$ to obtain the new model, formally,

$$\Theta' = \Theta - \alpha \nabla_\Theta \mathcal{L}_{tr}(\Theta), \tag{6}$$

where $\alpha$ is the updating rate. Note that, in the computation of class scores, since the reference node and query node can be both from the training set $\mathcal{V}_{tr}$, we set the computation weight $w_{i \to j} = 0$ if $i = j$ in Eqn. (3). After several iterates of gradient descent on $\mathcal{V}_{tr}$, we expect a better performance on the validation set $\mathcal{V}_{val}$, i.e., $\min_\Theta \mathcal{L}_{val}(\Theta')$. Thus, we can perform the gradient update as follows

$$\Theta = \Theta - \beta \nabla_\Theta \mathcal{L}_{val}(\Theta'), \tag{7}$$

where $\beta$ is the learning rate of meta optimization (Finn et al., 2017).

During the training process, we may perform batch sampling from training nodes and validation nodes, instead of taking all one time. In the testing stage, we may take all training nodes and perform the model update according to Eqn. (6) like the training process. The updated model is used as the final model and is then fed into Eqn. (2) to infer the class labels for those query nodes.

## 4 MODULES

In this section, we instantiate all modules (i.e., functions) of the aforementioned structure relation. The implementation details can be found in the following.

**Node Representation** $f_e(\mathcal{G}_{v_i})$: The local representation at vertex $v_i$ can be extracted by performing the graph convolution operation on subgraph $\mathcal{G}_{v_i}$. Similar to gridded images/videos, on which local convolution kernels are defined as multiple lattices with various receptive fields, the spectral graph convolution is used to encode the local representations of an input graph in our work.

Given a graph sample $\mathcal{G} = \{\mathcal{V}, \mathcal{E}, \mathcal{X}\}$, the normalized graph Laplacian matrix is $\mathbf{L} = \mathbf{I}_n - \mathcal{D}^{-1/2} \mathcal{E} \mathcal{D}^{-1/2} = \mathbf{U} \Lambda \mathbf{U}^T$, with a diagonal matrix of its eigenvalues $\Lambda$. The spectral graph convolution can be defined as the multiplication of signal $\mathcal{X}$ with a filter $g_\theta(\Lambda) = \mathrm{diag}(\theta)$ parameterized by $\theta$ in the Fourier domain: $\mathrm{conv}(\mathcal{X}) = g_\theta(\mathbf{L}) * \mathcal{X} = \mathbf{U} g_\theta(\Lambda) \mathbf{U}^T \mathcal{X}$, where parameter $\theta \in \mathbb{R}^n$ is a vector of Fourier coefficients. To reduce the computational complexity and obtain the local information, we use an approximate local filter of the Chebyshev polynomial (Defferrard et al., 2016), $g_\theta(\Lambda) = \sum_{k=0}^{K-1} \theta_k T_k(\hat{\Lambda})$, where parameter $\theta \in \mathbb{R}^K$ is a vector of Chebyshev coefficients and $T_k(\hat{\Lambda}) \in \mathbb{R}^{n \times n}$ is the Chebyshev polynomial of order $k$ evaluated at $\hat{\Lambda} = 2\Lambda/\lambda_{max} - \mathbf{I}_n$, a diagonal matrix of scaled eigenvalues. The graph filtering operation can then be expressed as $g_\theta(\Lambda) * \mathcal{X} = \sum_{k=0}^{K-1} \theta_k T_k(\hat{\mathbf{L}}) \mathcal{X}$, where $T_k(\hat{\mathbf{L}}) \in \mathbb{R}^{n \times n}$ is the Chebyshev polynomial of order $k$ evaluated at the scaled Laplacian $\hat{\mathbf{L}} = 2\mathbf{L}/\lambda_{max} - \mathbf{I}_n$. Further, we can construct multi-scale receptive fields for each vertex based on the Laplacian matrix $\mathbf{L}$, where each receptive field records hopping neighborhood relationships around the reference vertex $v_i$, and forms a local centralized subgraph.

**Path Reachability** $f_\mathcal{P}(v_i, v_j, \mathcal{E})$: Here we compute the probabilities of paths from vertex $i$ to vertex $j$ by employing random walks on graphs, which refers to traversing the graph from $v_i$ to $v_j$ according to the probability matrix $\mathbf{P}$. For the input graph $\mathcal{G}$ with $n$ vertices, the random-walk transition matrix

| Datasets | Nodes | Edges | Classes | Features | Label Rates |
|----------|-------|-------|---------|----------|-------------|
| Cora | 2,708 | 5,429 | 7 | 1,433 | 0.052 |
| Citeseer | 3,327 | 4,732 | 6 | 3,703 | 0.036 |
| Pubmed | 19,717 | 44,338 | 3 | 500 | 0.003 |
| NELL | 65,755 | 266,144 | 210 | 5,414 | 0.001 |

Table 1: The properties (especially for label rate) of various graph datasets used for the semi-supervised classification task.

can be defined as $\mathbf{P} = \mathcal{D}^{-1}\mathcal{E}$, where $\mathcal{D} \in \mathbb{R}^{n \times n}$ is the diagonal degree matrix with $\mathcal{D}_{ii} = \sum_i \mathcal{E}_{ij}$. That is to say, each element $P_{ij}$ is the probability of going from vertex $i$ to vertex $j$ in one step.

The sequence of nodes from vertex $i$ to vertex $j$ is a random walk on the graph, which can be modeled as a classical Markov chain by considering the set of graph vertices. To represent this formulation, we show that $P_{ij}^t$ is the probability of getting from vertex $v_i$ to vertex $v_j$ in $t$ steps. This fact is easily exhibited by considering a $t$-step path from vertex $v_i$ to vertex $v_j$ as first taking a single step to some vertex $h$, and then taking $t - 1$ steps to $v_j$. The transition probability $P^t$ in $t$ steps can be formulated as

$$P_{ij}^t = \begin{cases} P_{ij} & \text{if } t = 1 \\ \sum_h P_{ih} P_{h,j}^{t-1} & \text{if } t > 1 \end{cases}, \tag{8}$$

where each matrix entry $P_{ij}^t$ denotes the probability of starting at vertex $i$ and ending at vertex $j$ in $t$ steps. Finally, the node reachability from $v_i$ to $v_j$ can be written as a $d_p$-dimensional vector:

$$f_{\mathcal{P}}(v_i, v_j, \mathcal{E}) = [P_{ij}, P_{ij}^2, \dots, P_{ij}^{d_p}], \tag{9}$$

where $d_p$ refers to the step length of the longest path from $v_i$ to $v_j$.

**Class-to-Node Relationship** $s_{\mathcal{C}_c \to j}$: To define the node relationship $s_{i \to j}$ from $v_i$ to $v_j$, we simultaneously consider the property of path reachability $f_{\mathcal{P}}(v_i, v_j, \mathcal{E})$, local representations $f_e(\mathcal{G}_{v_i})$, and $f_e(\mathcal{G}_{v_j})$ of nodes $v_i, v_j$. The function $\phi_w(f_{\mathcal{P}}(v_i, v_j, \mathcal{E}))$ in Eqn. (3), which is to map the reachable vector $f_{\mathcal{P}}(v_i, v_j, \mathcal{E}) \in \mathbb{R}^{d_p}$ into a weight value, can be implemented with two 16-dimensional fully connected layers in our experiments. The computed value $w_{i \to j}$ can be further used to weight the local features at node $v_i$, $f_e(\mathcal{G}_{v_i}) \in \mathbb{R}^{d_v}$. For obtaining the similar score between $v_j$ and the $c$-th class nodes $\mathcal{C}_c$ in Eqn. (2), we perform a concatenation of two input features, where one refers to the weighted features of vertex $v_i$, and another is the local features of vertex $v_j$. One fully connected layer (w.r.t. $\phi_r$) with $C$-dimensions is finally adopted to obtain the relation regression score.

## 5 EXPERIMENTS

### 5.1 EXPERIMENTAL SETTINGS

We evaluate our proposed GIL method on three citation network datasets: Cora, Citeseer, Pubmed (Sen et al., 2008), and one knowledge graph NELL dataset (Carlson et al., 2010). The statistical properties of graph data are summarized in Table 1. Following the previous protocol in (Kipf & Welling, 2017; Zhuang & Ma, 2018), we split the graph data into a training set, a validation set, and a testing set. Taking into account the graph convolution and pooling modules, we may alternately stack them into a multi-layer Graph convolutional network. The GIL model consists of two graph convolution layers, each of which is followed by a mean-pooling layer, a class-to-node relationship regression module, and a final softmax layer. We have given the detailed configuration of the relationship regression module in the class-to-node relationship of Section 4. The parameter $d_p$ in Eqn. (9) is set to the mean length of between-node reachability paths in the input graph. The channels of the 1-st and 2-nd convolutional layers are set to 128 and 256, respectively. The scale of the respective filed is 2 in both convolutional layers. The dropout rate is set to 0.5 in the convolution and fully connected layers to avoid over-fitting, and the ReLU unit is leveraged as a nonlinear activation function. We pre-train our proposed GIL model for 200 iterations with the training set, where its initial learning rate, decay factor, and momentum are set to 0.05, 0.95, and 0.9, respectively. Here we train the GIL model using the stochastic gradient descent method with the batch size of 100. We further improve the inference learning capability of the GIL model for 1200 iterations with the validation set, where the meta-learning rates $\alpha$ and $\beta$ are both set to 0.001.

## 5.2 COMPARISON WITH STATE-OF-THE-ARTS

We compare the GIL approach with several state-of-the-art methods (Monti et al., 2017; Kipf & Welling, 2017; Zhou et al., 2004; Zhuang & Ma, 2018) over four graph datasets, including Cora, Citeseer, Pubmed, and NELL. The classification accuracies for all methods are reported in Table 2. Our proposed GIL can significantly outperform these graph Laplacian regularized methods on four graph datasets, including Deep walk (Zhou et al., 2004), modularity clustering (Brandes et al., 2008), Gaussian fields (Zhu et al., 2003), and graph embedding (Yang et al., 2016) methods. For example, we can achieve much higher performance than the deepwalk method (Zhou et al., 2004), e.g., 43.2% *vs* 74.1% on the Citeseer dataset, 65.3% *vs* 83.1% on the Pubmed dataset, and 58.1% *vs* 78.9% on the NELL dataset. We find that the graph embedding method (Yang et al., 2016), which has considered both label information and graph structure during sampling, can obtain lower accuracies than our proposed GIL by 9.4% on the Citeseer dataset and 10.5% on the Cora dataset, respectively. This indicates that our proposed GIL can better optimize structure relations and thus improve the network generalization. We further compare our proposed GIL with several existing deep graph embedding methods, including graph attention network (Velickovic et al., 2018), dual graph convolutional networks (Zhuang & Ma, 2018), topology adaptive graph convolutional networks (Du et al., 2017), Multi-scale graph convolution (Abu-El-Haija et al., 2018), etc. For example, our proposed GIL achieves a very large gain, e.g., 86.2% *vs* 83.3% (Du et al., 2017) on the Cora dataset, and 78.9% *vs* 66.0% (Kipf & Welling, 2017) on the NELL dataset. We evaluate our proposed GIL method on a large graph dataset with a lower label rate, which can significantly outperform existing baselines on the Pubmed dataset: 3.1% over DGCN (Zhuang & Ma, 2018), 4.1% over classic GCN (Kipf & Welling, 2017) and TAGCN (Du et al., 2017), 3.2% over AGNN (Thekumparampil et al., 2018), and 3.6% over N-GCN (Abu-El-Haija et al., 2018). It demonstrates that our proposed GIL performs very well on various graph datasets by building the graph inference learning process, where the limited label information and graph structures can be well employed in the predicted framework.

Table 2: Performance comparisons of semi-supervised classification methods.

| Methods | Cora | Citeseer | Pubmed | NELL |
|---|---|---|---|---|
| Clustering (Brandes et al., 2008) | 59.5 | 60.1 | 70.7 | 21.8 |
| DeepWalk (Zhou et al., 2004) | 67.2 | 43.2 | 65.3 | 58.1 |
| Gaussian (Zhu et al., 2003) | 68.0 | 45.3 | 63.0 | 26.5 |
| G-embedding (Yang et al., 2016) | 75.7 | 64.7 | 77.2 | 61.9 |
| DCNN (Atwood & Towsley, 2016) | 76.8 | - | 73.0 | - |
| GCN (Kipf & Welling, 2017) | 81.5 | 70.3 | 79.0 | 66.0 |
| MoNet (Monti et al., 2017) | 81.7 | - | 78.8 | - |
| N-GCN (Abu-El-Haija et al., 2018) | 83.0 | 72.2 | 79.5 | - |
| GAT (Velickovic et al., 2018) | 83.0 | 72.5 | 79.0 | - |
| AGNN (Thekumparampil et al., 2018) | 83.1 | 71.7 | 79.9 | - |
| TAGCN (Du et al., 2017) | 83.3 | 72.5 | 79.0 | - |
| DGCN (Zhuang & Ma, 2018) | 83.5 | 72.6 | 80.0 | 74.2 |
| Our GIL | **86.2** | **74.1** | **83.1** | **78.9** |

## 5.3 ANALYSIS

**Meta-optimization:** As can be seen in Table 3, we report the classification accuracies of semi-supervised classification with several variants of our proposed GIL and the classical GCN method (Kipf & Welling, 2017) when evaluating them on the Cora dataset. For analyzing the performance improvement of our proposed GIL with the graph inference learning process, we report the classification accuracies of GCN (Kipf & Welling, 2017) and our proposed GIL on the Cora dataset under two different situations, including "only learning with the training set $\mathcal{V}_{tr}$" and "with jointly learning on a training set $\mathcal{V}_{tr}$ and a validation set $\mathcal{V}_{val}$". "GCN /w jointly learning on $\mathcal{V}_{tr}$ & $\mathcal{V}_{val}$" achieves a better result than "GCN /w learning on $\mathcal{V}_{tr}$" by 3.6%, which demonstrates that the network performance can be improved by employing validation samples. When using structure relations, "GIL /w learning on $\mathcal{V}_{tr}$" obtains an improvement of 1.9% (over "GCN /w learning on $\mathcal{V}_{tr}$"), which can be attributed to the building connection between nodes. The meta-optimization strategy ("GIL /w meta-training from $\mathcal{V}_{tr} \rightarrow \mathcal{V}_{val}$" vs "GIL /w learning on $\mathcal{V}_{tr}$") has a gain of 2.9%, which indicates that a good inference capability can be learnt through meta-optimization. It is worth noting that, GIL adopts a meta-optimization strategy to learn the inference model, which is a process of migrating

from a training set to a validation set. In other words, the validation set is only used to teach the model itself how to transfer to unseen data. In contrast, the conventional methods often employ a validation set to tune parameters of a certain model of interest.

Table 3: Performance comparisons with several GIL variants and the classical GCN method on the Cora dataset.

| Methods | | Acc. (%) |
|---|---|---|
| GCN (Kipf & Welling, 2017) | /w learning on $\mathcal{V}_{tr}$ | 81.4 |
| | /w jointly learning on $\mathcal{V}_{tr}$ & $\mathcal{V}_{val}$ | 84.0 |
| GIL | /w learning on $\mathcal{V}_{tr}$ | 83.3 |
| | /w meta-train $\mathcal{V}_{tr} \rightarrow \mathcal{V}_{val}$ | **86.2** |
| GIL+mean pooling | /w 1 conv. layer | 84.5 |
| | /w 2 conv. layers | **86.2** |
| | /w 3 conv. layers | 85.4 |
| GIL+2 conv. layers | /w max-pooling | 85.2 |
| | /w mean pooling | **86.2** |

**Network settings:** We explore the effectiveness of our proposed GIL with the same mean pooling mechanism, but with different numbers of convolutional layers, i.e., "GIL + mean pooling" with one, two, and three convolutional layers, respectively. As can be seen in Table 3, the proposed GIL with two convolutional layers can obtain a better performance on the Cora data than the other two network settings (i.e., GIL with one or three convolutional layers). For example, the performance of 'GIL /w 1 conv. layer + mean pooling" is slightly decreased by 1.7% over "GIL /w 2 conv. layers + mean pooling" on the Cora dataset. Furthermore, we report the classification results of our proposed GIL by using mean and max-pooling mechanisms, respectively. GIL with mean pooling (i.e., "GIL /w 2 conv layers + mean pooling") can get a better result than the GIL method with max-pooling (i.e., "GIL /w 2 conv layers + max-pooling"), e.g., 86.2% vs 85.2% on the Cora graph dataset. The reason may be that the graph network with two convolutional layers and the mean pooling mechanism can obtain the optimal graph embeddings, but when increasing the network layers, more parameters of a certain graph model need to be optimized, which may lead to the over-fitting issue.

**Influence of different between-node steps:** We compare the classification performance within different between-node steps for our proposed GIL and GCN (Kipf & Welling, 2017), as illustrated in Fig. 2(a). The length of between-node steps can be computed with the shortest path between reference nodes and query nodes. When the step between nodes is smaller, both GIL and GCN methods can predict the category information for a small part of unlabeled nodes in the testing set. The reason may be that the node category information may be disturbed by its nearest neighboring nodes with different labels and fewer nodes are within 1 or 2 steps in the testing set. The GIL and GCN methods can infer the category information for a part of unlabeled nodes by adopting node attributes, when two nodes are not connected in the graph (i.e., step=$\infty$). By increasing the length of reachability path, the inference process of the GIL method would become difficult and more graph structure information may be employed in the predicted process. GIL can outperform the classic GCN by analyzing the accuracies within different between-node steps, which indicates that our proposed GIL has a better reference capability than GCN by using the meta-optimization mechanism from training nodes to validation nodes.

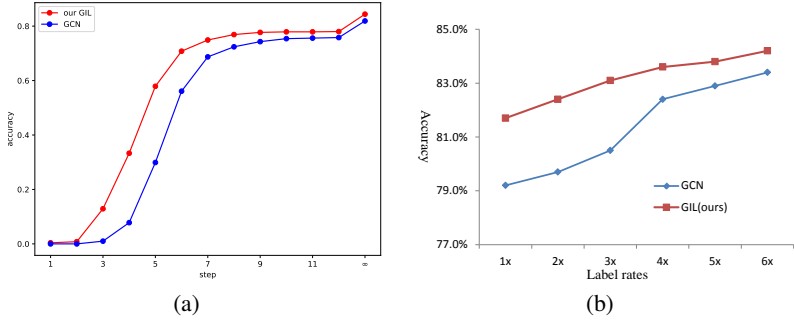

Figure 2: (a) Performance comparisons within different between-node steps on the Cora dataset. The accuracy equals to the number of correctly classified nodes divided by all testing samples, and is accumulated from step 1 to step $k$. (b) Performance comparisons with different label rates on the Pubmed dataset.

**Influence of different label rates:** We also explore the performance comparisons of the GIL method with different label rates, and the detailed results on the Pubmed dataset can be shown in Fig. 2(b). When label rates increase by multiplication, the performances of GIL and GCN are improved, but the relative gain becomes narrow. The reason is that, the reachable path lengths between unlabeled nodes and labeled nodes will be reduced with the increase of labeled nodes, which will weaken the effect of inference learning. In the extreme case, labels of unlabeled nodes could be determined by those neighbors with the $1 \sim 2$ step reachability. In summary, our proposed GIL method prefers small ratio labeled nodes on the semi-supervised node classification task.

**Inference learning process:** Classification errors of different epochs on the validation set of the Cora dataset can be illustrated in Fig. 3. Classification errors are rapidly decreasing as the number of iterations increases from the beginning to 400 iterations, while they are with a slow descent from 400 iterations to 1200 iterations. It demonstrates that the learned knowledge from the training samples can be transferred for inferring node category information from these reference labeled nodes. The performance of semi-supervised classification can be further increased by improving the generalized capability of the Graph CNN model.

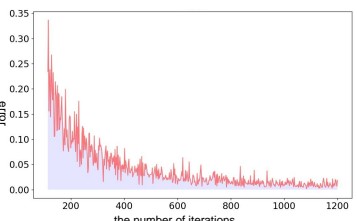

Figure 3: Classification errors of different iterations on the validation set of the Cora dataset.

Table 4: Performance comparisons with different modules on the Cora dataset, where $f_e$, $f_{\mathcal{P}}$, and $f_r$ denote node representation, path reachability, and structure relation, respectively.

| $f_e$ | $f_r$ | $f_{\mathcal{P}}$ | Acc.(%) |
|-------|-------|-------------------|---------|
| -     | -     | -                 | 56.0    |
| ✓     | -     | -                 | 81.5    |
| ✓     | ✓     | -                 | 85.0    |
| ✓     | ✓     | ✓                 | 86.2    |

**Module analysis:** We evaluate the effectiveness of different modules within our proposed GIL framework, including node representation $f_e$, path reachability $f_{\mathcal{P}}$, and structure relation $f_r$. Note that the last one $f_r$ defines on the former two ones, so we consider the cases in Table 4 by adding modules. When not using all modules, only original attributes of nodes are used to predict labels. The case of only using $f_e$ belongs to the GCN method, which can achieve 81.5% on the Cora dataset. The large gain of using the relation module $f_r$ (i.e., from 81.5% to 85.0%) may be contributed to the ability of inference learning on attributes as well as local topology structures which are implicitly encoded in $f_e$. The path information $f_{\mathcal{P}}$ can further boost the performance by 1.2%, e.g., 86.2% *vs* 85.0%. It demonstrates that three different modules of our method can improve the graph inference learning capability.

**Computational complexity:** For the computational complexity of our proposed GIL, the cost is mainly spent on the computations of node representation, between-node reachability, and class-to-node relationship, which are about $O((n_{tr} + n_{te}) * e * d_{in} * d_{out})$, $O((n_{tr} + n_{te}) * e * P)$, and $O(n_{tr} * n_{te} d_{out}^2)$, respectively. $n_{tr}$ and $n_{te}$ refer to the numbers of training and testing nodes, $d_{in}$ and $d_{out}$ denote the input and output dimensions of node representation, $e$ is about the average degree of graph node, and $P$ is the step length of node reachability. Compared with those classic Graph CNNs (Kipf & Welling, 2017), our proposed GIL has a slightly higher cost due to an extra inference learning process, but can complete the testing stage with several seconds on these benchmark datasets.

## 6 CONCLUSION

In this work, we tackled the semi-supervised node classification task with a graph inference learning method, which can better predict the categories of these unlabeled nodes in an end-to-end framework. We can build a structure relation for obtaining the connection between any two graph nodes, where node attributes, between-node paths, and graph structure information can be encapsulated together. For better capturing the transferable knowledge, our method further learns to transfer the mined knowledge from the training samples to the validation set, finally boosting the prediction accuracy of the labels of unlabeled nodes in the testing set. The extensive experimental results demonstrate the effectiveness of our proposed GIL for solving the semi-supervised learning problem, even in the few-shot paradigm. In the future, we would extend the graph inference method to handle more graph-related tasks, such as graph generation and social network analysis.

ACKNOWLEDGMENT

This work was supported by the National Natural Science Foundation of China (Nos. 61972204, 61906094, U1713208), the Natural Science Foundation of Jiangsu Province (Grant Nos. BK20191283 and BK20190019), and Tencent AI Lab Rhino-Bird Focused Research Program (No. JR201922).

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
