# OpenReview forum: "Graph inference learning for semi-supervised classification"
_ICLR.cc/2020/Conference — Accept (Poster)_

### Official Review · AnonReviewer1 · 2019-10-22
**Official Blind Review #1**

**Rating:** 6

**Review:**

This paper proposes to leverage the between-node-path information into the inference of conventional graph neural network methods. Specifically, the proposed method treats the nodes in training set as a reference corpus and, when infering the label of a specific node, makes this node "attend" to the reference corpus, where the "attention" weights are calculated based on the node representations and the between-node paths. (The paper used different terms about the "attention".)

In general, this paper shares similar insights like those in memory networks, which use a reference corpus to assist the inference on an individual sample. The major technical novelty of this paper seems to lie in the introduction of this idea on graphs and the use of between-node-path information for the "attention mechanism".

The paper is also well-organized and relatively easy to follow.

One of my major concerns is the unfair experimental comparison. The proposed method uses both the training set and validation set to train the model. And it's well-known that the offical split of the Cora, Citeseer, and Pubmed datasets has a validation set larger than the training set. As most of the baseline performances in Table 1 are obtained by using the training set only, the comparison of Table 1 is unfair and totally meaningless. A valid comparison is Table 2, where the baseline GCN is trained on both training set and validation set. But only the results on Cora is given and results on all other datasets are missing. Actually, I tried a quick run of GCN, without much tuning of hyper-parameters, on Citeseer with both training set and validation set as training data. And it can easily achieve a test accuracy of 0.75+, which is better than the performance of the proposed method reported in the paper.

Another concern is that, while the proposed method is intuitively sound, it is unclear how this method compares to naively increasing the number of layers of the GNN. Increasing the number of layers of the GNN can also eventually make the test node able to interact with all the training nodes, and the model should be able to implicitly distinguish the distances between nodes. Why incorporating between-node paths, and particularly in this way, helps?

Some minor points:
	- Could you clarify how the figure 2 is generated? By grouping the test nodes with differen between-nodes steps? It is confusing why performance on the nodes far away from the training nodes is better?
	- Why figure 3 is on PubMed while all other ablation analyses are done on Cora?

In summary, this paper shows some technical novelty but the motivation behind the proposed method is not strong enough. The experiment design is questionable and especially the main experiment comparison Table 1 is meaningless. Therefore I think this work is not ready for publish yet.

**Experience Assessment:**

I have published one or two papers in this area.

**Review Assessment: Checking Correctness Of Derivations And Theory:**

I carefully checked the derivations and theory.

**Review Assessment: Checking Correctness Of Experiments:**

I carefully checked the experiments.

**Review Assessment: Thoroughness In Paper Reading:**

I read the paper thoroughly.

---

### Official Review · AnonReviewer3 · 2019-10-22
**Official Blind Review #3**

**Rating:** 6

**Review:**

The paper suggests a learning architecture for graph-based semi-supervised learning. The input graph is given with only some labeled, and the goal is to label the rest. The architecture is trained on a validation set to learn features that optimize the accuracy of inference on the unlabeled node.

All parts of the architecture seem reasonable, and the experiments report advantage over prior work, which counts in favor of the paper. I found the writing to be somewhat dense and lack in clarity. My main concern is that I was unable to find a comparison of the techniques to prior work and an explanation of novelty. Unfortunately I am not very familiar with prior work on using neural networks for graph-based semi-supervised learning, and the lack of discussion made it difficult to put this work in context and assess its contribution. As of now I set my score to a weak reject, though I am willing to raise it if the positioning against prior work is explained and points to a substantial contribution.

A remark on clarity: it would help to use the appropriate commands for citations (citet and citep). The constant interruption of unbracketed references burdens the flow and readability of the text.

**Experience Assessment:**

I do not know much about this area.

**Review Assessment: Checking Correctness Of Derivations And Theory:**

I assessed the sensibility of the derivations and theory.

**Review Assessment: Checking Correctness Of Experiments:**

I assessed the sensibility of the experiments.

**Review Assessment: Thoroughness In Paper Reading:**

I read the paper at least twice and used my best judgement in assessing the paper.

---

### Official Review · AnonReviewer2 · 2019-10-26
**Official Blind Review #2**

**Rating:** 6

**Review:**

This paper presents a semi-supervised classification method for classifying unlabeled nodes in graph data. The authors propose a Graph Inference Learning (GIL) framework to learn node labels on graph topology. The node labeling is based of three aspects: 1) node representation to measure the similarity between the centralized subgraph around the unlabeled node and reference node; 2) structure relation that measures the similarity between node attributes; and 3) the reachability between unlabeled query node and reference node.

The authors propose to use graph convolution to learn the node representation and random work on graph to evaluate the reachability from query node to reference node.

The presentation of the paper is clear and easy to follow. The idea of the paper seems straightforward and the experimental results seems promising in semi-supervised classification for nodes in graph data.

Just a few concerns on model performance:

1) The node labeling is based on the comparison with reference nodes. Would such method get biased toward major classes in the data, if the data is imbalanced among different classes?

2) This method adopts three different modules for node labeling. It would be helpful if the authors can add some results to show the contribution of the different modules, i.e., what would be the performance if reachability or consistency of local graph topology structure is not considered in the classification?

One typo: in Table 1, it should be "Label rate".

**Experience Assessment:**

I do not know much about this area.

**Review Assessment: Checking Correctness Of Derivations And Theory:**

I did not assess the derivations or theory.

**Review Assessment: Checking Correctness Of Experiments:**

I assessed the sensibility of the experiments.

**Review Assessment: Thoroughness In Paper Reading:**

I read the paper at least twice and used my best judgement in assessing the paper.

---

### Decision · Program_Chairs · 2019-12-19

**Decision:**

Accept (Poster)

**Comment:**

The authors propose a graph inference learning framework to address the issues of sparse labeled data in graphs. The authors use structural information and node attributes to define a structure relation which is then use to infer unknown labels from known labels. The authors demonstrate the effectiveness of their approach on four benchmark datasets.

The approach presented in the paper is sound and the empirical results are convincing. All reviewers have given a positive rating for this paper. Two reviewers had some initial concerns about the paper but after the rebuttal they have acknowledged the answers given by the authors and adjusted their scores. R1 still has concerns about the motivation of the paper and I request the authors to adequately address this in their final version.